# The Use of Adaptive Sampling to Reach Disadvantaged Populations for Immunization Programs and Assessments: A Systematic Review

**DOI:** 10.3390/vaccines11020424

**Published:** 2023-02-13

**Authors:** Aybüke Koyuncu, Atsuyoshi Ishizumi, Danni Daniels, Mohamed F. Jalloh, Aaron S. Wallace, Dimitri Prybylski

**Affiliations:** Global Immunization Division, Centers for Disease Control and Prevention (CDC), Atlanta, GA 30329, USA

**Keywords:** adaptive sampling, vaccine-preventable diseases (VPDs), immunizations, vaccines, hard-to-reach populations

## Abstract

Vaccines prevent 4–5 million deaths every year, but inequities in vaccine coverage persist among key disadvantaged subpopulations. Under-immunized subpopulations (e.g., migrants, slum residents) may be consistently missed with conventional methods for estimating immunization coverage and assessing vaccination barriers. Adaptive sampling, such as respondent-driven sampling, may offer useful strategies for identifying and collecting data from these subpopulations that are often “hidden” or hard-to-reach. However, use of these adaptive sampling approaches in the field of global immunization has not been systematically documented. We searched PubMed, Scopus, and Embase databases to identify eligible studies published through November 2020 that used an adaptive sampling method to collect immunization-related data. From the eligible studies, we extracted relevant data on their objectives, setting and target population, and sampling methods. We categorized sampling methods and assessed their frequencies. Twenty-three studies met the inclusion criteria out of the 3069 articles screened for eligibility. Peer-driven sampling was the most frequently used adaptive sampling method (57%), followed by geospatial sampling (30%), venue-based sampling (17%), ethnographic mapping (9%), and compact segment sampling (9%). Sixty-one percent of studies were conducted in upper-middle-income or high-income countries. Data on immunization uptake were collected in 65% of studies, and data on knowledge and attitudes about immunizations were collected in 57% of studies. We found limited use of adaptive sampling methods in measuring immunization coverage and understanding determinants of vaccination uptake. The current under-utilization of adaptive sampling approaches leaves much room for improvement in how immunization programs calibrate their strategies to reach “hidden” subpopulations.

## 1. Introduction

Mortality and morbidity attributable to vaccine-preventable diseases (VPDs) have substantially declined over the past decades due to improvements in vaccination coverage globally, particularly in low- and middle-income countries (LMICs) [1,2]. However, 14 million children missed out on life-saving vaccines in 2019 alone [3]—pointing to inequities in vaccine coverage across and within countries. Some sub-populations may be consistently missed from immunization services due to structural inequities such as barriers to health care access (e.g., refugees), transient movement (e.g., nomads) or socioeconomic reasons (e.g., residents in slum settlements). Moreover, these sub-populations are often marginalized and stigmatized in the larger society, which then perpetuates their mistrust in authorities. Among refugee and immigrant communities, for example, mistrust of the healthcare system due to fears of deportation and perceived differential treatment negatively impact health-seeking behaviors [4]. The historical mistrust compounded over time with other geopolitical dynamics may lead to inequitable access and use of services by these sub-populations. In the context of immunization, “hidden” sub-populations and other hard-to-reach groups may not only be at high risk of being under-immunized, but also most likely to be missed from conventional methods for estimating immunization coverage that rely on official records (e.g., census) or household surveys as the basis for sampling frames [5,6].

Missing hidden populations may lead to overestimation of survey-based estimates of population-level immunization coverage, which may partly explain the continued occurrence of sizable outbreaks of VPDs in countries that purportedly have high childhood immunization coverage based on results from coverage surveys [7]. In Nigeria, targeted outreach activities aimed at nomadic and migratory populations in 2012 found that 52% of enumerated settlements in these populations had been historically excluded from microplans used to implement past polio supplementary immunization services such as mass immunization campaigns [8]. Isolated subpopulations that are unvaccinated can sustain transmission of VPDs even when surrounded by high population-level immunity in the general population [9]. Strategies to effectively reach zero-dose children (i.e., those who have never been vaccinated) and those with incomplete, delayed, or invalid vaccination are a key priority for the “last mile” of polio eradication and measles elimination efforts [9,10]. Disease eradication and elimination efforts are consequently threatened when certain subpopulations are consistently missed from vaccination services and left susceptible to VPDs and adverse health outcomes.

In 2018, the World Health Organization (WHO) updated their guidance on sampling strategies for assessing vaccination coverage in an effort to improve the rigor of conventional sampling methods [11]. Among other changes, the updated guidance suggests using probability sampling methods using census data and recommends using lists of enumeration areas to establish sampling frames. Although this guidance will undoubtedly improve the precision of survey results in comparison to non-probability sampling, the updated methods do not address potential selection bias due to the systematic underrepresentation of disadvantaged populations. Adaptive sampling methods, defined here as methods that modify or expand on conventional survey methods, may offer an alternative strategy for more effectively reaching population sub-groups. Adaptive sampling methods such as peer-driven sampling, for example, start with initial participants who are then tasked with recruiting their peers in a chain-referral method. Adaptive sampling methods have been used to estimate the size of hidden populations, understand their uptake of services, and inform strategies for delivering services to hard-to-reach [12] and “hidden” populations for HIV and sexually transmitted infection (STI) prevention [13,14]. Key subpopulations (e.g., sex workers, people who inject drugs, men who have sex with men) are at disproportionately greater risk for acquiring and transmitting HIV and other STIs than the general population [15]. Many individuals at high risk for HIV infection are socially marginalized due to their engagement in behaviors that are deemed illegal or illicit, and creating a sampling frame that sufficiently represents these groups using conventional approaches is infeasible or cost-prohibitive [13,14]. Despite the fact that these populations each represent a small proportion of the overall population size of any one particular country, they are important to target for population-based surveys [14,16,17]. 

Similarly to HIV and STIs, key subpopulations are at disproportionately greater risk of being under-immunized. Beyond implications for accurately estimating vaccination coverage, disadvantaged populations may also be missed in assessments examining the behavioral and social drivers of under-vaccination [18]. The duality of systemically missing sub-populations in immunization coverage surveys and socio-behavioral assessments may pose a serious threat to vaccine equity. Ensuring that the right to health is distributed equitably requires that “vaccines must be delivered to areas that are isolated geographically, culturally, socially or otherwise and to marginalized populations such as displaced people and migrants and those affected by conflict, political instability and natural disasters” (Immunization Agenda 2030) [19]. Successfully delivering immunizations to these groups, however, first requires a commitment to ensuring they are represented in assessments of immunization coverage and barriers to vaccine uptake. While the benefits of adaptive strategies have been documented in some public health fields, there is a lack of understanding regarding the potential utility of adaptive sampling approaches to reach hard-to-reach and hard-to-vaccinate subpopulations for immunization programs and assessments [12].

The SARS-CoV-2 pandemic is estimated to have put an additional 80 million children at increased risk of vaccine-preventable diseases due to disruptions in routine immunization services and delays in seeking services [3]. Despite the disruptions and delays, the pandemic has also illustrated the power of safe and effective immunizations in saving lives and protecting the human right to health [19]. Adaptive sampling approaches may be an important and necessary tool in the unprecedented global vaccination campaign to reach all individuals with COVID-19 vaccines. Here, we aim to review how adaptive sampling approaches have been used to estimate immunization coverage and/or assess the underlying behavioral and social drivers of vaccination uptake.

## 2. Materials and Methods

### 2.1. Search Strategy

We searched Scopus, PsychInfo, Medline, Embase, Cochrane Library, and CINAHL to identify relevant studies regardless of published language and published through November 2020. Search terms included keywords related to immunization (e.g., “vaccin*”, “immuni*”) and keywords associated with adaptive sampling. Adaptive sampling keywords consisted of terms used to describe common adaptive sampling methods (e.g., “respondent-driven”, “geospatial”) and terms related to study populations typically targeted by adaptive sampling (e.g., “transient”, “hard-to-reach”) (see Appendix A for a full list of search terms). The search terms were identified and refined based on a review of key references and discussion among co-authors, with assistance from a reference librarian at the US Centers for Disease Control and Prevention (US CDC). We uploaded search results to covidence (www.covidence.org), which we used to manage the subsequent steps of the review.

### 2.2. Study Selection

For both stages of study selection (abstract review, full text review), two independent reviewers (A.K and A.I) screened each article for eligibility. All discrepancies between reviewers were resolved by discussion and consensus. 

Studies were included if they met the following inclusion criteria: (i) described research on any vaccines (defined as substances used to stimulate the production of antibodies and provide immunity against one or several diseases) or immunization programs; (ii) described the use of adaptive sampling approaches to identify individuals for service delivery or participation in primary data collection; adaptive sampling was defined as substantive modifications to or expansions of conventional survey methods (with probability or non-probability sampling) to reach population sub-groups that would otherwise have a high likelihood of being missed; (iii) conducted among humans; (iv) published in English, French, or Spanish. 

Studies were excluded if they met any of the following criteria: (i) abstract or full text not available; (ii) duplicative studies; (iii) described immunologic surveys without assessing immunization coverage; (iv) not published in a peer-reviewed journal (e.g., dissertations, conference proceedings). Studies that used adaptive methods (e.g., geospatial mapping) solely to improve immunization systems without conducting any sampling or primary data collection from respondents [20,21] were excluded.

### 2.3. Data Extraction

We extracted data from the full text of all included studies using a data extraction form developed by the authors. Data extraction for all included studies was conducted independently by two reviewers (A.K and A.I), and any discrepancies in data extracted were resolved by discussion and consensus. 

We extracted data on population type, eligibility criteria for the target population, data collection type (i.e., quantitative, qualitative, or both), vaccine-preventable disease of interest, sampling methods, study objectives, and whether immunization services were delivered as part of the study. We categorized studies that collected data on vaccines generally without specifying a vaccine-preventable disease as collecting data related to all immunizations generally. We categorized sampling methods into one of the following groups: peer-driven sampling (e.g., respondent-driven sampling, snowball sampling, etc.), geospatial sampling (e.g., using satellite imagery to identify housing structures and/or populations), compact segment sampling [22], ethnographic mapping [6], and venue-based sampling (e.g., time–location sampling). Categories for sampling methods were developed iteratively based on the data extracted during the literature review.

Studies that utilized peer-driven sampling were further categorized as being partially or fully compliant with respondent-driven sampling approaches. Studies were categorized as partially compliant with respondent-driven sampling approaches if they mentioned the term “respondent-driven sampling” and at least one of the following methodological attributes: (i) use of coupons or referral management systems; (ii) use of seeds; and/or (iii) use of specialized analytic software for respondent-driven sampling (e.g., Respondent Driven Sampling Analyst (RDSA)). Respondent-driven sampling is a specialized form of peer-driven sampling that uses a formal mechanism for tracking referrals and adjusting for network size in data analysis to calculate analytic weights that qualify the method as a probability sampling method (as opposed to non-probability sampling). Studies that utilized peer-driven sampling were, therefore, only categorized as fully compliant with respondent-driven sampling approaches if all of the above methodological attributes were included in the manuscript. 

We extracted the study objectives listed in each manuscript and utilized inductive thematic analysis [23] to allow the extracted data to guide our selection of major themes. We then categorized study objectives based on six major themes: measuring coverage; measuring attitudes, behaviors and vaccine uptake barriers; delivering vaccines; enhancing microplanning; health systems improvement; and comparing sampling methods. In addition, we generated three separate binary categorical variables (0 = no; 1 = yes) to indicate whether a study collected and presented data on (i) knowledge and attitudes about immunizations (knowledge on availability of services, perceived need for immunizations, etc.); (ii) barriers to vaccination uptake or reasons for under-vaccination; and/or (iii) vaccination uptake. For this initial systematic review, we noted only whether the data were present or absent in the manuscript and did not extract specific data presented in each manuscript. Studies were categorized as having collected data on immunization uptake if they collected data on immunization status based on self-report, immunization cards, or serological testing. We created a categorical variable to classify studies as having collected immunization-related data from adults about themselves, from mothers/caregivers about their children, or both. For studies using quantitative data collection methods, we created additional categorical variables indicative of whether the study examined predictors of immunization knowledge/attitudes and/or uptake.

### 2.4. Data Analysis

We performed descriptive analyses to examine the variability in study characteristics, focusing on the following areas: type of adaptive sampling method; reason for using adaptive sampling; type of study population; antigen or vaccine-preventable disease of focus; type of data collection method (quantitative or qualitative); type of data presented. We categorized countries by income level using the 2021 World Bank classifications [24]. Statistical analyses were conducted in Stata 16 (College Station, TX, USA).

## 3. Results

After the exclusion of duplicates, our search identified 3069 studies for abstract screening, of which 132 met the criteria for full-text review. We identified 23 studies [25,26,27,28,29,30,31,32,33,34,35,36,37,38,39,40,41,42,43,44,45,46,47] that satisfied the inclusion criteria after full-text review (Figure 1).

The 23 studies were conducted in 19 countries from 1987 to 2017 (Table 1). 

Studies included a mix of qualitative (30%; N = 7) and quantitative (70%; N = 16) data collection methods and no mixed-methods studies were identified. We identified more studies in upper-middle-income or high-income countries (61%; N = 14) compared to lower-middle-income or low-income countries (39%; N = 9) [15].

A majority of studies identified in this review (65%) did not focus on key disadvantaged subpopulations that may be more likely to be missed in conventional sampling approaches, such as sex workers, migratory populations, refugees, persons living in high-density urban slums, or undocumented immigrants. Adaptive sampling methods were used to collect data from a variety of target populations including the general population (26%), mothers and caregivers of children (39%), healthcare providers (17%), sex workers (13%), and other groups (35%) such as refugees, migrant populations, immigrants/ethnic minorities, internally displaced persons (IDPs), prisoners, and men who have sex with men. Immunization-related data were most often collected from adults about themselves (35%; N = 8), from mothers and caregivers about their children (35%; N = 8), or both (13%; N = 3). Among all antigens, studies most frequently collected immunization-related data on hepatitis B specifically (26%) and all immunizations generally (26%). Stratified by income, studies conducted in lower-middle-income or low-income countries most frequently collected data on cholera (33%) or measles (33%), while studies in upper-middle-income or high-income countries most frequently collected data on hepatitis B (43%). 

### 3.1. Sampling Methods

Peer-driven sampling was the most frequently used adaptive sampling method (N = 13; 57%), followed by geospatial sampling (30%), venue-based sampling (17%), ethnographic mapping (9%), and compact-segment sampling (9%). Five studies utilized multiple sampling methods [26,29,44,46,47]. Four studies compared geospatial sampling or compact-segment sampling methods to conventional approaches on metrics such as population size (i.e., number of households identified), immunization coverage estimates, cost, time, amount of advanced preparation and expertise needed, design effect, rate of homogeneity, and sociodemographic characteristics of identified populations [36,42,45,46]. The advantages and disadvantages of adaptive strategies varied. For example, Milligan et al. and Gong et al. found no meaningful differences in vaccination coverage between compact-segment sampling methods and/or GIS sampling and conventional EPI sampling methods [45,46]. In contrast, Barau et al. identified over 3000 settlements that were previously not included in microplans for polio vaccination campaigns in Nigeria when using satellite imagery to identify households compared to hand-drawn maps [36]. 

Only three studies used adaptive sampling methods to deliver immunization services. Puga et al. and Magalhaes et al. used respondent-driven sampling methods to deliver hepatitis B vaccines to female sex workers in Brazil by recruiting an initial sample of sex workers from prostitution environments (e.g., night clubs, brothels) and incentivizing them to recruit their peers [31,39]. Barau et al. utilized geospatial sampling to provide polio vaccines to children in Nigeria by using high-resolution satellite imagery to identify settlements for vaccination teams to visit [36]. Two studies utilized adaptive sampling as part of outbreak prevention and response activities. Specifically, geospatial sampling was used to identify households in response to cholera outbreaks in South Sudan [33] and Zambia [43].

Among 13 of 23 studies that used peer-driven sampling [25,28,30,31,32,34,35,37,39,40,41,44,47], only 2 studies were partially compliant with accepted respondent-driven sampling methods [31,35], and one study was fully compliant with respondent-driven sampling methods [39]. All three studies that were partially or fully compliant with respondent-driven sampling methods were focused on hepatitis B and collected data or delivered services to female sex workers [31,39] or other adults aged 16 and above [35]. Studies that used peer-driven sampling but were not compliant with respondent-driven sampling methods primarily utilized snowball sampling methods in which study participants referred their peers, but no formal mechanism of tracking referrals or adjusting for network size in data analysis was used.

Geospatial methods included random spatial sampling to select data collection starting points [33,38,42,43,47] for the use of satellite imagery to identify potential residential structures and conduct grid-based sampling [46] and the use of geographic information systems (GIS) to generate maps for data enumerators and use GPS tracking to follow enumerators [36]. Among studies that used venue-based sampling (4 of 23), two studies sampled only based on location [27,44], and two studies utilized sampling based on both time and location [26,29]. Coreil et al. and Gullion et al. utilized venue-based sampling to collect data on routine childhood immunizations from mothers/caregivers identified at Mother’s clubs, gatherings of roadside sellers and neighbors, at rally posts delivering child health services in Haiti [27] and other venues including but not limited to schools, bookstores, chiropractors, health food stores, holistic health centers, and yoga classes in the United States [44]. Baars et al. utilized time–location sampling at bars and brothels to collect data and deliver hepatitis B vaccination services to female sex workers and men who have sex with men [26,29]. 

### 3.2. Types of Data Collected

Data on immunization uptake were collected in 65% of studies (N = 15) [26,29,31,32,33,34,35,38,39,40,42,43,45,46,47], while data on knowledge and attitudes about immunizations were collected in 57% of studies (N = 13) [25,26,27,28,29,30,31,32,34,38,40,43,44]. Among studies that collected data on immunization uptake (N = 15), 27% aimed to assess coverage following antigen-specific supplementary immunization campaigns (SIAs) for cholera or meningitis [33,38,42,43]. Fifty-two percent of studies specifically collected data on reasons for non-vaccination or barriers to vaccination (N = 12) [25,26,27,29,32,33,34,38,40,43,44,47]. Among studies that collected quantitative data (16 of 23), six studies analyzed predictors of immunization-related knowledge and attitudes (38%) and nine studies analyzed predictors of immunization uptake (56%).

## 4. Discussion

We identified 23 studies in the published literature that utilized adaptive sampling strategies to collect quantitative and qualitative data for immunization coverage assessments, microplanning for vaccine delivery, VPD outbreak response, and behavioral assessments of barriers to vaccine uptake. Adaptive sampling strategies were more frequently used in high-income settings, and few studies collected data from populations known to be hard-to-reach. Among the 13 studies that utilized peer-driven approaches, only one study incorporated methodological attributes that provided a probability sample. This review shows that the current use of adaptive sampling methods in the immunization literature is limited, and opportunities exist to improve immunization programs by re-orienting the use of adaptive sampling approaches to find and learn about hidden populations that are disproportionately affected by VPDs.

The use of adaptive sampling in other health sectors can act as a roadmap for the potential utility of these approaches in supplementing conventional immunization surveys. Population-based approaches alone were not sufficient to control the HIV/AIDS epidemic, and over time, the need for interventions focused on marginalized high-risk groups became apparent. The WHO Biobehavioral survey guidelines for populations at risk for HIV aims to address these challenges by providing a menu of sampling strategies including conventional and adaptive approaches along with decision-making criteria to select sampling methods [48]. Factors considered in the selection of sampling methods include whether there are existing sampling frames, the size of the target population, whether the population is socially networked, and whether the population is accessible at physical locations [48]. Similarly, for immunizations, efforts for improving coverage have historically utilized a population-based approach in defined geographic areas rather than focusing on subpopulations at high risk of under-immunization. Despite global progress in expanding immunization coverage, gains in coverage have stalled short of targets in many countries due to under-vaccinated subpopulations [12,49,50]. Immunization Agenda 2030 outlines a shift in strategy to prioritize global health equity and reach zero-dose children, which will likely necessitate adaptive sampling methods highlighted in this review. 

Adaptive approaches warrant consideration for supplementing conventional approaches, particularly in settings where key differences in uptake, knowledge, attitudes, and behaviors may exist between those that would be included and excluded by conventional methods as well as in settings in which in-depth behavioral insights are needed to address barriers to vaccination [51]. While for HIV/STIs, adaptive strategies are used to sample adults in high-risk groups, our findings illustrate the feasibility of using adaptive strategies for sampling mothers and caregivers in order to reach immunization-eligible children. Our findings also demonstrate that the added value of adaptive approaches relative to conventional sampling approaches is context-specific. The WHO Behavioral and Social Drivers of Vaccination (BeSD) guidebook includes some guidance on developing a sampling plan and the advantages and disadvantages of adaptive sampling strategies such as snowball sampling when trying to understand reasons for low vaccine uptake [52]. Additional studies are needed to help researchers and immunization program mangers assess whether adaptive sampling strategies are needed in their context. Even among the 13 studies identified in this review that utilized peer-driven sampling, there was methodological heterogeneity, and few studies were compliant with accepted respondent-driven sampling methods. Standardized guidance on how to use adaptive methodologies to achieve various goals of immunization programs (e.g., coverage surveys) will be critical for achieving the global health equity agenda. 

The COVID-19 pandemic underscores how increasing global coverage of SARS-CoV-2 vaccines is not only an ethical obligation but also critical for preventing the emergence and spread of harmful SARS-CoV-2 variants in the global landscape [53]. Adaptive sampling strategies are among the many tools that can be used to ensure equitable immunization coverage for vaccines against SARS-CoV-2 as well as routine immunizations. The feasibility of adaptive sampling approaches for reaching a variety of disadvantaged populations has been documented in many settings [36,46,54,55,56]. While adaptive strategies are cost-effective in some settings, cost-effectiveness can vary by specific sampling strategy, target population, and setting. Additional research is needed to evaluate the utility and feasibility of these approaches in strengthening routine immunization systems as well as VPD outbreak response activities. 

This review has several limitations. We did not conduct any risk of bias assessments and did not exclude studies on the basis of whether adaptive sampling methods were used appropriately. We likely underestimated the number of qualitative studies related to immunizations that have utilized adaptive sampling methodologies such as snowball sampling, given that these methods are more common in qualitative research and may, therefore, have been excluded from the Methods sections of published articles.

## 5. Conclusions

Our systematic review revealed that there is a paucity of studies in the literature that have used adaptive sampling approaches to collect data related to immunizations. Although the studies we identified used adaptive sampling strategies for various purposes, many of them were limited to high-income settings or did not have a specific focus on disadvantaged populations. Taken together, our findings highlight the overarching need to more proactively explore the potential effectiveness and practicability of applying adaptive sampling strategies in ensuring equitable immunization coverage. The current under-utilization of adaptive sampling approaches leaves much room for improvement in how immunization programs calibrate their strategies to reach disadvantaged populations that may be recurrently missed in immunization assessments and service delivery.

## Figures and Tables

**Figure 1 vaccines-11-00424-f001:**
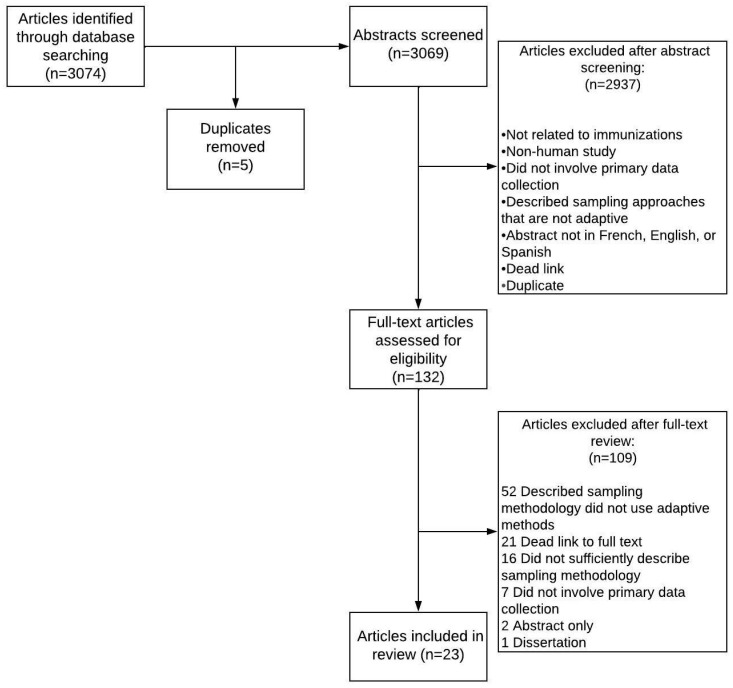
Flow diagram for study screening and selection.

**Table 1 vaccines-11-00424-t001:** Characteristics of included studies and major themes of adaptive sampling use.

Study	Year	Country	Sampling Type	Target Population	Age Eligibility	Vaccines ^A^	Study Objectives
Milligan, P. et al. [45]	2000	GMB	Compact segment	Mothers and caregivers of children	Mothers and caregivers of children aged 12–23 months	Routine immunizations; yellow fever	Measure coverage; Compare sampling methods
Baars, J. E. et al. [29]	2004	NLD	Ethnographic mapping; Venue-based	Men who have sex with men	Not specified	Hepatitis B	Measure coverage; Measure attitudes, behaviors and vaccine uptake barriers
Baars, J. E. et al. [26]	2004	NLD	Ethnographic mapping; Venue-based	Sex workers	Not specified	Hepatitis B	Measure coverage; Measure attitudes, behaviors and vaccine uptake barriers
Grais, R. F. et al. [42]	2006	NER	Geospatial	General population	Not specified	Meningitis	Measure coverage; Compare sampling methods
Massing, L. A. et al. [38]	2014	DRC	Geospatial	General population	≥1 year	Oral cholera	Measure coverage; Measure attitudes, behaviors and vaccine uptake barriers
Ferreras, E. et al. [43]	2016	ZMB	Geospatial	General population	>1 year	Oral cholera	Measure coverage; Measure attitudes, behaviors and vaccine uptake barriers
Parker, L. A. et al. [33]	2015	SSD	Geospatial	General population; internally displaced persons; slum residents; healthcare providers; prisoners/inmates	>1 year	Oral cholera	Measure coverage; Measure attitudes, behaviors and vaccine uptake barriers
Barau, I. et al. [36]	2012	NGA	Geospatial	Mothers and caregivers of children	Not specified	Oral polio	Enhance microplanning; Deliver vaccines; Compare sampling methods
Gong, W. et al. [46]	2016	PAK	Geospatial; Compact segment	Mothers and caregivers of children	Mothers and caregivers of children aged 12–23 months	Routine immunizations	Measure coverage; Compare sampling methods
Shand, L. et al. [34]	2008	AUS	Peer-driven	General population	18–26 years	Human papillomavirus	Measure coverage; Measure attitudes, behaviors and vaccine uptake barriers
Peel, R. et al. [41]	2016	AUS	Peer-driven	General population	55–60 years	Pneumococcal polysaccharide	Compare sampling methods
Pearce, C. et al. [30]	2003	AUS	Peer-driven	Healthcare providers	Not specified	Hepatitis B; All immunizations generally	Measure attitudes, behaviors and vaccine uptake barriers
Russell, G. et al. [28]	2001	CAN	Peer-driven	Healthcare providers	Not specified	Influenza	Measure attitudes, behaviors and vaccine uptake barriers
Koehlmoos, T. P. et al. [37]	Unknown	BGD	Peer-driven	Healthcare providers; government staff; health systems staff that do not directly provide care	Not specified	All immunizations generally; Routine immunizations	Health systems improvement
Jama, A. et al. [32]	2013	SWE	Peer-driven	Mothers and caregivers of children; immigrants/ethnic minority	Mothers and caregivers of children aged 18 months to 5 years	Routine immunizations	Measure coverage; Measure attitudes, behaviors and vaccine uptake barriers
Ben Natan, M. et al. [25]	2016	ISR	Peer-driven	Mothers and caregivers of children	Not specified	Pertussis	Measure attitudes, behaviors and vaccine uptake barriers
McDonald, P. et al. [40]	2017	USA	Peer-driven	Mothers and caregivers of children	Mothers and caregivers of children in kindergarten (~5 years) through 12th grade (~17–18 years)	All immunizations generally	Measure coverage; Measure attitudes, behaviors and vaccine uptake barriers
Puga, M. A. M. et al. [39]	2009	BRA	Peer-driven and fully compliant with RDS	Sex workers	Not specified	Hepatitis B	Measure coverage; Deliver vaccines; Measure attitudes, behaviors and vaccine uptake barriers
Yang, Y. et al. [35]	2008	CHN	Peer-driven and partially compliant with RDS	Migrants	Not specified	Hepatitis B	Measure coverage; Measure attitudes, behaviors and vaccine uptake barriers
Magalhaes, R.L.B et al. [31]	2014	BRA	Peer-driven and partially compliant with RDS	Sex workers	≥18 years	Hepatitis B	Measure coverage; Measure attitudes, behaviors and vaccine uptake barriers; Deliver services
Roberton, T. et al. [47]	2014	JOR; LBN	Peer-driven; Geospatial	Mothers and caregivers of children; Refugees	Mothers and caregivers of children aged 12–23 months	All immunizations generally; Routine immunizations	Measure coverage; Measure attitudes, behaviors and vaccine uptake barriers
Gullion, J. S. et al. [44]	Unknown	USA	Peer-driven; Venue-based	Mothers and caregivers of children	Not specified	All immunizations generally	Measure attitudes, behaviors and vaccine uptake barriers
Coreil, J. et al. [27]	1987	HTI	Venue-based	Mothers and caregivers of children	Mothers and caregivers of children aged 12–23 months	All immunizations generally	Measure attitudes, behaviors and vaccine uptake barriers

^A^ Routine immunizations include childhood vaccines to protect against hepatitis, polio (oral polio vaccine), diphtheria, pertussis, tetanus, *Haemophilus influenzae* type B, measles, mumps, rubella, and tuberculosis (Bacillus Calmette-Guerin).

## Data Availability

The data presented in this study are available on request from the corresponding author.

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
