# Peer review of "The Use of Adaptive Sampling to Reach Disadvantaged Populations for Immunization Programs and Assessments: A Systematic Review"

_vaccines, 2023, doi:10.3390/vaccines11020424_

Round 1

Reviewer 1 Report

This article deals with a very important topic, namely how to reach disadvantaged populations when it comes to immunization and to which extent are adaptive sampling techniques used in this field. Often low immunization rates are related  to characteristics of the populations or the individuals (e.g. concepts like  hesitancy or choice), but I agree with the authors that professionals and scientists need to rethink their strategies of reach, and thus also their sampling techniques.

My rating scores are low, but that means that I think the article can be improved.  I am not an expert in systematic reviews, but I have the feeling that the review is carried out well. However, the text, the presentation of background, results and discussion is quite 'technical' and provides more ''technical information' than understanding. It is more an inventory than an understanding of studies that use adaptive sampling. After having read it, I still felt that I did not fully understand it. I will give more specific comments and suggestions to improve the paper.

1. In the introduction quite some claims are made that are not backed up or embedded in  literature. ''Missing hidden populations may ...''(51) ....Beyond implications  .. may also be missed ....''(64-66) ... ''alternatives are not well understood'' (69)

It is concluded that there is lack of understanding of the potential utility of adaptive sampling .... (77). I would say, adaptive sampling is not often used in the field of immunization. Then this can be understood in different ways. That question - how to understand the under-utilisation of adaptive sampling -  can be addressed in the results, as it may be addressed in the articles that are reviewed, or it can be addressed by the authors in the discussion.  

2. Categorizations of sampling. Ok as such, but where does it come from? it is based on the literature, a conventional categorization, or the result of the analysis? Also, these need to be explained a bit more - but that can be done in the results, where concrete examples can be given, to stimulate  contextual understanding of these sampling techniques by the reader. 

 3. Results. This is too much of a list with information, an inventory, and not so much an analysis. The reader needs a better description of the sampling techniques: what do these mean? the reader needs more concrete examples.  (peer driven??? geospatial? compact segment sampling? etc)

And the results as presented now articulate a kind of arbitrary ness. The  raises which  patterns are seen in the reviews? Associations between sampling techniques, specific populations/target groups, in specific regions? or other patterns? Part of these patterns are mentioned in the discussion (high/low income settings), but I think these are results of the analysis. 

4. In the discussion many things are addressed that can/need to be dealt with in the introduction, as they explain the rationale of the study more then that they discuss the results in the context of the broader literature on adaptive sampling (like WHO policies, the right to health versus under serving specific populations, the fruitful use of adaptive sampling in other fields). What are current understandings of the use of these sampling strategies in other fields? is under use related to the quantitative ideals of evidence in public health? to the standardization of research in this field? to the disconnection with social sciences approaches? to the costs of these techniques? What are  explanations in the literature and how do these apply to under use of these methods in immunization?,  

So, while I think that the paper may be important, I think it can do more to present the insights in a good way and to help readers to learn from the study. 

Author Response

Dear Reviewer, 

Thank you for your thoughtful feedback, which we believe has greatly improved our manuscript. We have incorporated your suggestions and have responded to each of your comments below. 

Sincerely,

Aybüke Koyuncu, on behalf of all co-authors

  1. In the introduction quite some claims are made that are not backed up or embedded in literature. ''Missing hidden populations may ...''(51) ....Beyond implications .. may also be missed ....''(64-66) ... ''alternatives are not well understood'' (69)

Thank you for flagging this, the appropriate references have been added (Updated lines 52-55 and 85-86). Line 69 has been removed as it is covered in other sections of the introduction.

It is concluded that there is lack of understanding of the potential utility of adaptive sampling .... (77). I would say, adaptive sampling is not often used in the field of immunization. Then this can be understood in different ways. That question - how to understand the underutilisation of adaptive sampling - can be addressed in the results, as it may be addressed in the articles that are reviewed, or it can be addressed by the authors in the discussion.

Prior to conducting this systematic review, it was not clear how and how frequently adaptive sampling was used in the field of immunization. The objective of this review was not to understand underutilization of adaptive sampling but instead to understand how adaptive sampling approaches have been used to estimate immunization coverage and/or assess the underlying behavioral and social drivers of vaccination uptake (Updated lines 125-127).

  1. Categorizations of sampling. Ok as such, but where does it come from? it is based on the literature, a conventional categorization, or the result of the analysis? Also, these need to be explained a bit more - but that can be done in the results, where concrete examples can be given, to stimulate contextual understanding of these sampling techniques by the reader.

Thank you for the opportunity to clarify. The categories were developed iteratively based on the data extracted in the literature review. We have added text to clarify this in the methods section (Updated lines 180-182).  In order to provide more concrete examples of each sampling method we have added some illustrative examples in section 3.1.

  1. Results. This is too much of a list with information, an inventory, and not so much an analysis. The reader needs a better description of the sampling techniques: what do these mean? The reader needs more concrete examples. (peer driven???geospatial? compact segment sampling? etc) And the results as presented now articulate a kind of arbitraryness. The raises which patterns are seen in the reviews? Associations between sampling techniques, specific populations/target groups, in specific regions? or other patterns? Part of these patterns are mentioned in the discussion (high/low income settings), but I think these are results of the analysis.

Thank you for this feedback. Given that only 23 papers were identified in this review, we did not have sufficient data to explore patterns and associations.

Regarding your feedback on the various sampling techniques, to help the reader understand the sampling techniques:

  • Peer-driven sampling is defined in the introduction (Updated lines 69-71)
  • We have provided references in the methods section that the reader can use if they are interested in learning more about compact segment sampling and ethnographic mapping (Updated lines 179-180)
  • We added text in the methods to provide an example of geospatial sampling (Updated lines 178-179)
  • We added additional illustrative examples of respondent driven sampling and geospatial sampling methods in section 3.1 (Updated lines 295-299).
  • Illustrative examples of geospatial sampling and venue-based sampling are included in section 3.1 (Updated lines 313-341)

  1. In the discussion many things are addressed that can/need to be dealt with in the introduction, as they explain the rationale of the study more then that they discuss the results in the context of the broader literature on adaptive sampling (like WHO policies, the right to health versus under serving specific populations, the fruitful use of adaptive sampling in other fields). What are current understandings of the use of these sampling strategies in other fields? is under use related to the quantitative ideals of evidence in public health? to the standardization of research in this field? to the disconnection with social sciences approaches? to the costs of these techniques? What are explanations in the literature and how do these apply to under use of these methods in immunization?

Thank you for the suggestion, we have shifted the text on the right to health and health equity as well as the utility of adaptive sampling in other fields to the introduction to help motivate the rational of our study (Updated lines 74-118).

The scope of our review was limited to understanding if and how adaptive sampling strategies are being used in immunization literature rather than why the methods are or are not being used. However, to address your feedback we have added some text to the discussion to hypothesize why adaptive strategies have not been frequently used in immunization research (Updated lines 408-417)

We have also added text and references in the discussion that clarifies that the feasibility of these methods has been documented in various settings for various target populations, but that the cost-effectiveness of these methods varies based on the sampling strategy used, target population, and setting (Updated lines 435-440)

Reviewer 2 Report

The authors undertake a review of peer reviewed papers about immunisation coverage and vaccination uptake, with a focus on identifying papers that use adaptive sampling methods as opposed to conventional probability based sampling methods. Of over 3000 papers, only 23 papers have the detail they require. That in itself indicates that adaptive sampling methods are very rarely used. Of those, only 22% include as their focus a hard-to-reach sub-population. Clearly, there is little use of adaptive sampling methods. The discussion mentions that even the WHO guidance on sampling strategies doesn't mention adaptive methodologies, which may explain why they are not used. 

your paper may be strengthened by showing how adaptive sampling methods have provided improvements in other jurisdictions

Specific: 

Lines 265-267: can you please strengthen the results part that shows evidence for this statement. I take it you are talking about the 'other groups' at line 203, but it isn't clear. If you include sex workers, then it is 35%. By the way, do these papers that use adaptive methodologies provide results that help the immunisation of the hard-to-reach sub populations?

Line 269: I would start the sentence with 'This review shows that the current use...'

At the end of the paragraph from line 273 to 291 I have the impression that the updated guidelines created by WHO do not allow adaptive approaches. Then by the end of the next paragraph I am confused, as you seem to indicate that they are options available in the guidelines for adaptive sampling. Are you saying that they are not being sufficiently encouraged for surveys related to immunisation?

There is a word missing between 'that' and 'be' on line 312

I think you could be stronger in your recommendation provided at line 314, such as adaptive sampling approaches have been shown to be cost-effective in other jurisdictions (with references). There are several available, for instance, see the special issue of Methodological Innovations Online regarding survey sampling for hard-to-reach populations (2010)

Author Response

Dear Reviewer, 

Thank you for your thoughtful feedback, which we believe has greatly improved our manuscript. We have incorporated your suggestions and have responded to each of your comments below. 

Sincerely,

Aybüke Koyuncu, on behalf of all co-authors

Lines 265-267: can you please strengthen the results part that’s hows evidence for this statement. I take it you are talking about the 'other groups' at line 203, but it isn't clear. If you include sex workers, then it is 35%. By the way, do these papers that use adaptive methodologies provide results that help the immunization of the hard-to-reach sub populations?

Thank you for raising this point. We have updated the text to clarify that a majority of studies (65%) did not collect data on key disadvantaged subpopulations including sex workers and the other groups (e.g. refugees) captured in the “other groups” category.

The results of these papers can help improve immunization coverage in hard-to-reach sub populations in a variety of ways based on the type of data collected. For example, data on knowledge and attitudes about vaccinations in a hard-to-reach population can help understand social and behavioral barriers to uptake and help inform community engagement strategies. Adaptive sampling strategies can be useful not only for collecting data but also for delivering interventions. Although only three papers identified in this review provided interventions (Puga et al. and Magalhaes et al. used respondent-driven sampling methods to deliver hepatitis B vaccines to female sex workers in Brazil (25,33) and Barau et al. utilized geospatial sampling to provide polio vaccines to children in Nigeria (30)) these articles are a useful proof of concept for the feasibility of using adaptive sampling to deliver immunizations.

Line 269: I would start the sentence with 'This review shows that the current use...'

Thank you for the suggestion, the sentence has been updated accordingly.

At the end of the paragraph from line 273 to 291 I have the impression that the updated guidelines created by WHO do not allow adaptive approaches. Then by the end of the next paragraph am confused, as you seem to indicate that they are options available in the guidelines for adaptive sampling. Are you saying that they are not being sufficiently encouraged for surveys related to immunisation?

Thank you for the opportunity to clarify. The updated WHO guidelines do not include any consideration of adaptive sampling approaches. We have added language that clarifies that adaptive strategies could help address some of the limitations in the existing guidance, but that they are not currently included in the guidance. 

There is a word missing between 'that' and 'be' on line 312

This missing word has been added.

I think you could be stronger in your recommendation provided at line 314, such as adaptive sampling approaches have been shown to be cost-effective in other jurisdictions (with references). There are several available, for instance, see the special issue of Methodological Innovations Online regarding survey sampling for hard-to-reach populations (2010)

Thank you for this suggestion. We have strengthened the recommendation by including references that demonstrate the feasibility and effectiveness of adaptive sampling strategies in other research fields (Updated lines 435-440)

Upon a review of the literature, adaptive sampling strategies appear to be cost-effective in some settings but not in others. Therefore, we have updated the text with the following:

While adaptive strategies are cost-effective in some settings, cost-effectiveness can vary by specific sampling strategy, target population, and setting.” (Updated lines 436-438).

Round 2

Reviewer 1 Report

Thank you for the second version of this article. I think it has improved but I still feel that the results and the discussion can be improved more.

The first paragraph of the discussion provides information that is actually ''results'' . Also the table contains information about sampling strategy and target group, that I would expect to be analyzed and addressed in the results.  One would expect that researchers chose a sampling method for a reason, but the reader does not learn anything about that: how do researchers  consider the rationale and effectiveness of their method considering their aim and targetgroup, and region? How do they reflect on their study? as the sample is not big, one would expect analysis of the content of the articles. The results is still mostly a list of information, is very procedural,  and not an analysis of the content of the reviewed papers. 

Also, it is only in the discussion the reader learns about the fact that the WHO had developed standards for  sampling strategies. I think that needs to be put in the introduction, as that is part of the landscape, background and rationale of the study.  In the discussion one would expect that the questions is discussed how the reviewed papers relate to these policies? And how does the analysis of all these papers - the complete sample -  relate to/inform these policies? What do we actually learn from this review? what are new insights? what is the added value considering what we already know about adaptive sampling? The discussion is rather general and it appears not to be related to the content of the results. 

Author Response

Dear Reviewer, 

Thank you for your time spent improving this article. We have responded to your feedback below.  

Sincerely,

Aybüke Koyuncu, on behalf of all co-authors

The first paragraph of the discussion provides information that is actually ''results'' .

Thank you for bringing this to our attention. We have shifted some of the paragraph to the results section (Updated lines 229-232). The updated first paragraph synthesizes findings that are already presented in the results section.

Also the table contains information about sampling strategy and target group, that I would expect to be analyzed and addressed in the results. One would expect that researchers chose a sampling method for a reason, but the reader does not learn anything about that: how do researchers consider the rationale and effectiveness of their method considering their aim and target group, and region? How do they reflect on their study? as the sample is not big, one would expect analysis of the content of the articles. The results is still mostly a list of information, is very procedural, and not an analysis of the content of the reviewed papers.

The objective of this review was to understand how adaptive sampling approaches have been used to estimate immunization coverage and/or assess the underlying behavioral and social drivers of vaccination uptake. An assessment as to determine whether the methods used were appropriate is outside the scope of this paper and cannot be conducted the papers identified in this review did not consistently include justification for their selected methods. The results therefore focus instead on synthesizing how the sampling methods were used, with illustrative examples provided throughout.

To address your feedback, we have added the following to the results section:

  • Updated lines 241-244 provide an analysis of trends in antigens across country income: “Stratified by income, studies conducted in lower-middle-income or low-income countries most frequently collected data on cholera (33%) or measles (33%) while studies in upper-middle-income or high-income countries most frequently collected data on hepatitis B (43%).”
  • Updated lines 254-260 provide more illustrative examples from the papers identified in this review: “The advantages and disadvantages of adaptive strategies varied. For example, Milligan et al. and Gong et al. found no meaningful differences in vaccination coverage between compact-segment sampling methods and/or GIS sampling and conventional EPI sampling methods (45,46). In contrast, Barau et al. identified over 3000 settlements that were previously not included in microplans for polio vaccination campaigns in Nigeria when using satellite imagery to identify households compared to hand-drawn maps (36).”
  • Updated lines 300-302 provide an illustrative example of why studies collected data on immunization uptake: “Among studies that collected data on immunization uptake (N=15), 27% aimed to assess coverage following antigen-specific supplementary immunization campaigns (SIAs) for outbreak-prone pathogens such as cholera or meningitis (33,38,42,43)”

Also, it is only in the discussion the reader learns about the fact that the WHO had developed standards for sampling strategies. I think that needs to be put in the introduction, as that is part of the landscape, background and rationale of the study.

Thank you for this suggestion, we have shifted the background on the WHO guidance on sampling strategies to the introduction (Updated lines 66-73).

In the discussion one would expect that the questions is discussed how the reviewed papers relate to these policies? And how does the analysis of all these papers - the complete sample – relate to/inform these policies? What do we actually learn from this review? what are new insights? what is the added value considering what we already know about adaptive sampling? The discussion is rather general and it appears not to be related to the content of the results.

A main finding of this review is that adaptive sampling strategies are currently not being used widely in immunization programs and assessments. We therefore utilize the discussion section to expand on why adaptive strategies were historically needed in other public health fields, their demonstrated feasibility in many settings, and why it is critical that these strategies should be considered and used more widely in immunization programs.

In order to address your feedback, we have added additional points reflecting on other key findings of the review:

  • Updated lines 419-422: “While for HIV/STIs adaptive strategies are used to sample adults in high-risk groups, our findings illustrate the feasibility of using adaptive strategies to sample mothers and caregivers in order to reach immunization-eligible children.”

  • Updates lines 422-431: “Our findings also demonstrate that the added value of adaptive approaches relative to conventional sampling approaches is context specific…Additional studies are needed to help researchers and immunization program mangers assess whether adaptive sampling strategies are needed in their context. Even among the 13 studies identified in this review that utilized peer-driven sampling, there was methodological heterogeneity and few studies were compliant with accepted respondent-driven sampling methods.“

Round 3

Reviewer 1 Report

I think the article has improved substantially and the meaning of the results are communicated much more clearly now the relationship with other studies is articulated better.